# Identification and Characterization of a Novel Splice Site Mutation Associated with Glycogen Storage Disease Type VI in Two Unrelated Turkish Families

**DOI:** 10.3390/diagnostics11030500

**Published:** 2021-03-12

**Authors:** Sarah C. Grünert, Luciana Hannibal, Anke Schumann, Stefanie Rosenbaum-Fabian, Stefanie Beck-Wödl, Tobias B. Haack, Mona Grimmel, Miriam Bertrand, Ute Spiekerkoetter

**Affiliations:** 1Department of General Paediatrics, Adolescent Medicine and Neonatology, Faculty of Medicine, Medical Centre University of Freiburg, 79106 Freiburg, Germany; anke.schumann@uniklinik-freiburg.de (A.S.); stefanie.rosenbaum-fabian@uniklinik-freiburg.de (S.R.-F.); ute.spiekerkoetter@uniklinik-freiburg.de (U.S.); 2Laboratory of Clinical Biochemistry and Metabolism, Department of General Paediatrics, Adolescent Medicine and Neonatology, Faculty of Medicine, Medical Centre-University of Freiburg, 79106 Freiburg, Germany; luciana.hannibal@uniklinik-freiburg.de; 3Institute of Medical Genetics and Applied Genomics, University of Tübingen, 72076 Tübingen, Germany; Stefanie.Beck-Woedl@med.uni-tuebingen.de (S.B.-W.); Tobias.Haack@med.uni-tuebingen.de (T.B.H.); mona.grimmmel@med.uni-tuebingen.de (M.G.); miriam.bertrand@med.uni-tuebingen.de (M.B.); 4Center for Rare Diseases, University of Tübingen, 72076 Tübingen, Germany

**Keywords:** glycogen metabolism, splice variant, glycogen phosphorylase, PYGL, transcriptome analysis, in silico analysis

## Abstract

Introduction: Glycogen storage disease type VI (GSD VI) is a disorder of glycogen metabolism due to mutations in the *PYGL* gene. Patients with GSD VI usually present with hepatomegaly, recurrent hypoglycemia, and short stature. Results: We report on two non-related Turkish patients with a novel homozygous splice site variant, c.345G>A, which was shown to lead to exon 2 skipping of the PYGL-mRNA by exome and transcriptome analysis. According to an in silico analysis, deletion Arg82_Gln115del is predicted to impair protein stability and possibly AMP binding. Conclusion: GSD VI is a possibly underdiagnosed disorder, and in the era of next generation sequencing, more and more patients with variants of unknown significance in the *PYGL*-gene will be identified. Techniques, such as transcriptome analysis, are important tools to confirm the pathogenicity and to determine therapeutic measures based on genetic results.

## 1. Introduction

Glycogen storage disease type VI (GSD VI, OMIM #232700) is a disorder of glycogen metabolism due to mutations in the *PYGL* gene resulting in deficiency of hepatic glycogen phosphorylase (PYGL). Different isoforms of glycogen phosphorylase are expressed in various tissues including brain (PYGB), heart (PYGB), muscle (PYGM), and liver (PYGL). In contrast to the intra-organ activity of brain and muscle isoforms of glycogen phosphorylase, liver PYGL allows rapid release of free glucose into circulation, thus providing a constant supply of energy to extrahepatic tissues [1]. It is also the rate-limiting enzyme of glycogen degradation.

GSD VI affects approximately 1:65,000–1:85,000 live births [2]. The disorder is clinically characterized by hepatomegaly, poor growth, and short stature [2]. Typical laboratory findings include ketotic hypoglycemia, elevated hepatic transaminases, hyperlipidemia, and reduced prealbumin concentrations. Hepatic fibrosis is a common finding in GSD VI, while cirrhosis and hypertrophic cardiomyopathy are rare complications [2]. Clinical and biochemical abnormalities may decrease with age, but the risk of hypoglycemia and ketosis may persist. Only about 50 patients have been described in the literature so far [3]. GSD VI is considered a relatively mild disorder that presents in infancy and childhood [2], however, some severe cases with recurrent hypoglycemia and marked hepatomegaly have also been described [4].

Some individuals with GSD VI may not require any treatment, others significantly benefit from a high-protein diet (2–3 g/kg/day) with frequent small meals and supplementation of uncooked cornstarch to improve growth. Reduction of total carbohydrates and especially simple sugars is recommended to reduce glycogen storage in the liver [2]. Suboptimal metabolic control may result in short stature, delayed puberty, osteopenia/osteoporosis, and hepatic fibrosis. Although the tumor risk is low, hepatic adenomas and hepatocellular carcinoma can develop [5,6].

We herein describe two non-related Turkish patients with a novel homozygous splice site variant that was shown to lead to exon 2 skipping of the PYGL-mRNA by exome and transcriptome analysis. According to in silico analysis, deletion Arg82_Gln115del is predicted to impair AMP-mediated activation.

## 2. Materials and Methods

### 2.1. Sample Collection and RNA Isolation

Blood was collected in a PAXgene^TM^ Blood RNA Tube (Qiagen, England Biolabs, MA, USA) using a standard blood collection set. Total RNA was extracted with the QIAsymphony RNA Kit (Qiagen, England Biolabs, MA, USA) and RNA purification was done using RNeasy technology (silica membrane).

### 2.2. Trio Exome Sequencing

Coding genomic regions were enriched with a SureSelect^XT^ Human All Exon Kit V7 (Agilent Technologies, Santa Clara, CA, USA) for subsequent sequencing on a NovaSeq 6000 (Illumina, San Diego, CA, USA).

### 2.3. Transcriptome Sequencing of PAXgene Blood

RNA quality was assessed with the Agilent 2100 Fragment Analyzer total RNA kit (Agilent Technologies, Inc., Santa Clara, CA, USA). The sample had high RNA integrity number (RIN > 9). Using the NEBNext Ultra II Directional RNA Library Prep kit with 100 ng of total RNA input for each sequencing library, poly(A)-selected sequencing libraries were generated according to the manufacturer’s manual. All libraries were sequenced on the Illumina NovaSeq 6000 platform (Illumina, San Diego, CA, USA) as 2 × 100 bp paired-end reads and to a depth of approximately 50 million clusters each. Quality of raw RNA-seq data in FASTQ files was assessed using Read QC (version 2020_03, https://github.com/imgag/ngs-bits, (accessed on 15 December 2020)) to identify potential sequencing cycles with low average quality and base distribution bias. Reads were preprocessed with SeqPurge (version 2020_03, https://github.com/imgag/ngs-bits, (accessed on 15 December 2020)) and aligned using STAR (version 2.7.3a, https://github.com/alexdobin/STAR/, (accessed on 15 December 2020)) allowing spliced read alignment to the human reference genome (build GRCh37).

### 2.4. Analysis of PYGL Protein Structure

The X-ray crystal structure of human liver PYGL was downloaded from the Protein Data Bank under accession number 1FA9 (PYGL complexed with AMP, pyridoxal phosphate, and alpha-D-glucopyranose). The complete description of PYGL structure 1FA9 can be found in its associated publication [7]. Images of human recombinant PYGL highlighting amino acid residues lost due to the mutation were created with PyMOL for Mac (PyMOL™ version 2.4.0, Schrodinger, LLC).

## 3. Results

### 3.1. Case Presentations

#### 3.1.1. Patient 1

The female patient is the first child of Turkish parents. The family history is unremarkable for metabolic disorders. Pregnancy and birth were uneventful. The girl was partially breastfed until age 2.5 years. She showed normal growth until 6 months of age. At that age, failure to thrive occurred and the body length dropped below the 3rd centile. At the age of 3, elevated transaminase activities were first noted and the patient was referred to the pediatric gastroenterology department. Clinical investigation was unremarkable apart from mild hepatomegaly (liver 2–3 cm below costal margin, ultrasound: liver diameter about 11 cm) and failure to thrive. Her body length was on the 0.2nd centile, and body weight on the 7th centile. Transaminase activities were elevated (AST 105 U/L, normal < 35 U/L; ALT 156 U/L, normal < 45 U/L), while the serum triglyceride concentration was at the upper limit of the reference range (166 mg/dL, normal 30–150 mg/dL). Glycogen storage disease type Ia was ruled out by mutation analysis, and liver biopsy was performed at the age of 3 years and 10 months revealing glycogen accumulation with mild fibrosis and steatosis. Phosphorylase and phosphorylase B kinase activities were decreased in liver tissue suggestive of GSD VI. The diagnosis was confirmed by Sanger mutation analysis in *PYGL,* which yielded a novel homozygous variant affecting the last base of exon 2, c.345G>A. The mutation was classified as variant of unknown significance as the base exchange itself would be associated with a silent mutation (p.Q115=). However, due to the position of the variant, effects on splicing were considered likely. Blood glucose monitoring showed mild hypoglycemia after an overnight fast and the patient was put on a diet with frequent meals. The further clinical course was uncomplicated with short stature and elevated transaminase activities remaining the major clinical features. Lactate concentrations were found to be slightly elevated at several occasions (maximum 3.6 mmol/L). At 10 years of age, a protein-rich diet was started, which has led to a slight catch-up growth. Body length at age 14 years is however still below the 3rd centile. The concentration of AST normalized (31 U/L, normal < 35 U/L), the ALT is only mildly elevated (50 U/L, normal < 35 U/L). Her overall clinical condition is excellent.

#### 3.1.2. Patient 2

The girl, first child of consanguineous Turkish parents, first presented at age 13 months with hepatomegaly and short stature (body length < 1st centile, body weight 18th centile). Further diagnostic work-up revealed elevated transaminase activities (AST 65 U/L, ALT 123 U/L, normal < 35 U/L) and severe hypertriglyceridemia (triglycerides 1406 mg/dl, normal < 150 mg/dL). Uric acid, creatine kinase, total cholesterol, and coagulation parameters were normal. Liver sonography showed distinct hepatomegaly, but no splenomegaly. Intensive investigations to rule out infectious diseases (toxoplasma, EBV, CMV, parvovirus B19, HHV6, hepatitis A, B, and C), alpha-1 antitrypsin deficiency, mucopolysaccharidoses and other metabolic diseases were performed and yielded negative results. A blood glucose and lactate profile for 30 h showed only one mild episode of hypoglycemia of 61 mg/dL with lactate levels ranging from 1.6 to 6.7 mmol/L. Clinical symptoms of hypoglycemia were not reported by the parents. The psychomotor development was normal. Trio exome sequencing revealed homozygosity for the same variant of unknown significance as in patient 1, c.345G>A.

After the diagnosis was made, another blood glucose and ketone profile was made which showed mild hypoglycemia during the night (minimum 62 mg/dL) with elevated ketone levels up to 1.7 mmol/L, while glucose levels during the day were stable > 80 mg/dL. The girl was put on a protein-rich diet with mild reduction of carbohydrates. Additionally, two doses of cornstarch (1 g/kg) were added at bedtime and at 3 am at night. Under this treatment, no hypoglycemia was documented, and transaminase activities as well as triglyceride concentrations almost normalized within 3 months (AST 44 U/L, ALT 37 U/L, normal < 35 U/L; triglycerides 175 mg/dL, normal < 150 mg/dL). The girl also showed some catch-up growth (3 cm within 3 months).

### 3.2. Exome and Transcriptome Analysis in Patient 2

Trio exome analysis of patient 2 and both parents yielded a novel homozygous splice site variant in *PYGL*, c.345G>A. This variant was predicted to change the last nucleotide of exon 2, resulting in loss of the splice donor site of exon 2 of the *PYGL* gene. On protein level this results in an in-frame deletion of 34 amino acids corresponding with deletion of exon 2 in the main transcript. As the pathogenicity of this variant was unclear transcriptome analysis was performed. This technique indeed confirmed a homozygous loss of exon 2 in the PYGL-mRNA and a consequently reduced activity of hepatic phosphorylase (Figure 1). According to HGVS: *PYGL*(ENST00000216392):c.[244_356del]; [244_356del], p.[(Arg82_Gln115del)]; [(Arg82_Gln115del)].

### 3.3. Predicted Effect of the Homozygous p.Arg82_Gln115del Mutation on the Enzymatic Activity of PYGL

Human liver glycogen phosphorylase (PYGL) is a homodimer formed by monomers of 846 amino acid residues. The activity of the enzyme is controlled by phosphorylation of a highly conserved serine residue (Ser14) [1]. The homodimer possesses two regions, namely, a regulatory region that contains the phosphorylation peptide harboring Ser14 (amino acids 5–22) and AMP, and a catalytic region located on the opposite side of the protein that binds to the carbohydrate substrate and the cofactor pyridoxal phosphate (PLP) (Figure 2A, PDB accession code 1FA9) [7]. The binding site for the allosteric regulator AMP is located in helix-2 of the protein structure spanning amino acid residues 48–78 (colored in dark blue in Figure 2A–C, with Tyr75 providing stabilizing bonding interactions [7]. The deletion identified in the patients comprises amino acids 82 to 115 (Figure 2A–C, colored in yellow), hence only three amino acid residues away from the essential AMP binding site. The large structural loss caused by deletion of residues Arg82 to Gln115 is likely to disrupt protein stability, and possibly AMP binding, thereby impairing the transition of PYGL from its inactive to active states via allosteric modulation. AMP acts by reducing the K_M_ of the enzyme for its substrate glucose-1-phosphate [1]. While the deleted amino acids are not located within or near the catalytic site in the primary structure, the conformational contacts and motions that are lost between the regulatory and catalytic domains could conceivably impact catalysis. 

## 4. Discussion

GSD type VI is a rare inborn error of glycogen metabolism with only about 50 cases reported so far. The severity of clinical symptoms varies significantly. While most patients present with hepatomegaly [2,3,6], the risk of hypoglycemia is lower than compared to most other types of hepatic GSDs [2]. This was also true for our two patients, and ketotic hypoglycemia only occurred after prolonged fasting in early childhood. However, both our patients showed severe failure to thrive/short stature. In a study by Szymanska et al. including 16 patients with ketotic hepatic GSDs (type III/VI/IX,) short stature was the most common complication during the long-term follow-up in this patient cohort. Unfortunately, the authors did not distinguish between the different subtypes of ketotic GSDs, but short stature was present in 31% of cases in the whole cohort [8]. In contrast, Aeppli et al., who recently reported on the long-term outcome of six GSD VI patients, found that dietary treatment was successful in all cases to normalize growth and development [3]. Although some patients do not require treatment for hypoglycemia, most have better growth with therapy [2]. Growth hormone therapy should be avoided because it usually exacerbates ketosis and may increase the risk of complications [2].

While hyperlactatemia is a typical feature of GSD type I and is associated with impaired growth, it is no common finding in patients with ketotic forms of hepatic GSDs [9]. Interestingly, both our patients showed elevated lactate levels, at least intermittently. Beauchamp et al. described elevated lactate concentrations in six GSD VI patients, of which some only showed postprandial lactate elevations, while in others, lactate was elevated both pre- and postprandially [4]. In the patients reported by Beauchamp et al., hyperlactatemia was usually associated with severe and recurrent hypoglycemia and severe hepatomegaly [4]. 

The genetic background of GSD VI is heterogeneous, and apart from the Mennonite pathogenic variant c.1620+1G>A that generates a transcript lacking all or part of exon 13 while maintaining some residual enzyme activity, no common mutations have been identified so far [10]. Therefore, it is interesting that we found the same homozygous splice site mutation in two unrelated Turkish families. Besides the Mennonite variant c.1620+1G>A that is known to be associated with a milder course of the disease, no clear genotype-phenotype correlations exist in GSD VI [2]. Although in the majority of patients deficiency of liver glycogen phosphorylase is the result of missense mutations that either affect substrate binding, pyridoxal phosphate binding or activation of glycogen phosphorylase, several splice site mutations have been reported [11]. For the novel splice site mutation c.244_356del identified in our patients we could show by transcriptome analysis that this variant leads to loss of exon 2 of the PYGL-mRNA confirming its pathogenicity. In silico analysis suggests deleterious effects on protein stability and possibly on the activation mediated by AMP binding to PYGL and possible reduction of enzymatic activity. In vitro studies with purified human recombinant PYGL missing amino acid 82 to 115 would be helpful to determine the degree of residual activity of this pathogenic variant.

## 5. Conclusions

As GSD VI is often associated with rather mild symptoms, it is well conceivable that this disorder is underdiagnosed. In the era of next generation sequencing, it can be assumed that more and more patients with variants of unknown significance in the *PYGL*-gene will be identified by these diagnostic approaches. Therefore, techniques such as transcriptome analysis are important tools to confirm the pathogenicity and to determine therapeutic measures based on the genetic results.

## Figures and Tables

**Figure 1 diagnostics-11-00500-f001:**
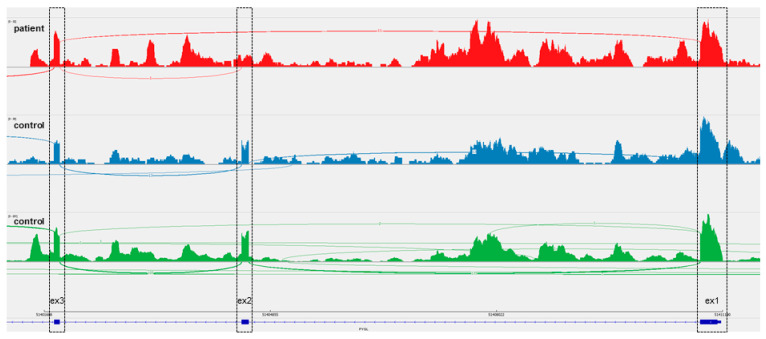
Transcriptome analysis indicates skipping of exon 2 in *PYGL* mRNA. Results of RNA sequencing of PAXgene blood are shown as a Sashimi plot for patient (red) and two controls (blue and green). Exon 1 (ex1, right) and 3 (ex3, left) show comparable expression for all samples, exon 2 (ex2, middle) is skipped in patient RNA of *PYGL* (exon numbering and exon/intron structure displayed is according to ENST00000216392.7).

**Figure 2 diagnostics-11-00500-f002:**
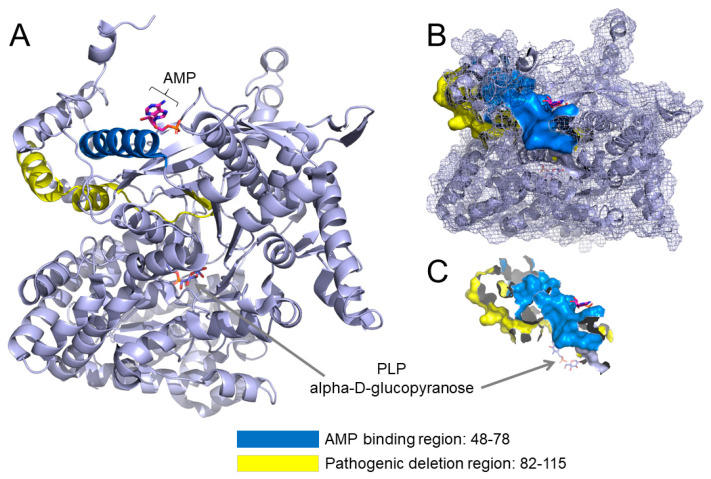
Structure of the liver isoform of human glycogen phosphorylase. (**A**) PYGL monomer displaying the AMP binding site in helix-2 (ribbons, dark blue), AMP (sticks representation, magenta) and the deletion mutant comprising residues Arg82 to Gln115 (ribbons, yellow) in the regulatory region of the protein. Cofactor pyridoxal phosphate (PLP) and inhibitor alpha-D-glucopyranose are shown as orange sticks within the catalytic region of the protein. (**B**) Overlay of ribbons and mesh representations depicting the spatial arrangement of the protein with the AMP binding site shown in dark blue (surface representation) and deletion fragment colored yellow (surface representation). (**C**) Close up view of the nearly contiguous AMP binding site (amino acid residues 48 to 78) and deleted region Arg82_Gln115. The X-ray structure of human hepatic PYGL was downloaded from the Protein Data Bank, accession number 1FA9 and described in [7].

## Data Availability

No new data were created or analyzed in this study. Data sharing is not applicable to this article.

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
