# Peer review of "Identification and Characterization of a Novel Splice Site Mutation Associated with Glycogen Storage Disease Type VI in Two Unrelated Turkish Families"

_diagnostics, 2021, doi:10.3390/diagnostics11030500_

Round 1

Reviewer 1 Report

This manuscript presents a short description of newly identified splice site mutation in the PYGL gene, leading to glycogen storage disease type VI. The effect of the mutation has been confirmed by RNA-seq analysis. Possible changes at the protein level were predicted by molecular modeling. This is an interesting and well performed study. Nevertheless, some minor revision is recommended.

  1. Abstract – The last sentence is very general, and it is not derived from results presented in this study. It is trivial, at best, and should be removed. The same is valid to the last sentence of Conclusions.
  2. Line 42 – References is required to cite works which demonstrated severe cases of the disease.
  3. Materials and Methods – Description of the biological material is absent, while it is necessary to shown what cells/tissues were analyzed, and how was RNA isolated and purified.
  4. The first paragraph of Discussion mostly repeats the information provided in Introduction. This redundancy should be eliminated.

Author Response

We thank the reviewer for his/her positive assessment and valuable suggestions.

The reviewer’s comments have been addressed as follows:

  1. We agree with the reviewer and have deleted the last sentence of the abstract. Nevertheless, we would like to mention it in the conclusion as our case shows the importance of such techniques to better characterize patients with variants of unknown significance.
  2. We have added Beauchaump et al. 2007 as a reference here.
  3. We have added information on blood sampling and RNA isolation:

“Sample collection and RNA isolation

Blood was collected in a PAXgeneTM Blood RNA Tube (Qiagen, England Biolabs) uing a standard blood collection set. Total RNA was extracted with the QIAsymphony RNA Kit (Qiagen, England Biolabs) and RNA purification was done using RNeasy technology (silica membrane).”

  1. We agree with the reviewer and have tried to eliminate redundancy.

Reviewer 2 Report

Dear Authors,

congratulations for pointing out the attention on this rvery rare disorders and to the world of the transcriptome analysis in the era of NGS. We appreciate also the well detailed paper withb beautiful figures and captures.

There are some point to dicuss:

  • did you perform glucagon test in these patients?
  • could you describe better the fasting tolerance?
  • In the title you mention "unrelated Turkish families" but in the text in patient 2 you wrote consanguineous parents. Could you clarify it?
  • Which did you intend for "stamina"?
  • At which age hepatomegaly began?

Author Response

We thank the reviewer for this very positive evaluation.

1) Although it might be helpful in some cases, glucagon tests have not been performed in our patients.

2) Unfortunately, there is no good blood glucose documentation, especially in patient 1. Therefore, the exact fasting tolerance cannot be evaluated.

3) This might be misleading. What was meant is that the two families are not related to each other, although they are both of Turkish origin. The parents of patient 2, however, were consanguineous.

4) We have deleted this word.

5) In patient 1, this is unfortunately not quite clear. At first presentation to our gastroenterologic department, it was already present (information given in case presentation 1). In patient 2, it was first noted at 13 months.